# Diagnostic Value of Bronchoscopy for Peripheral Metastatic Lung Tumors

**DOI:** 10.3390/cancers14020375

**Published:** 2022-01-13

**Authors:** Yoshie Tsujimoto, Yuji Matsumoto, Midori Tanaka, Tatsuya Imabayashi, Keigo Uchimura, Takaaki Tsuchida

**Affiliations:** 1Department of Endoscopy, Respiratory Endoscopy Division, National Cancer Center Hospital, Tokyo 1040045, Japan; ytsujimoto@hosp.ncgm.go.jp (Y.T.); midotana@ncc.go.jp (M.T.); timabaya@ncc.go.jp (T.I.); kuchimur@ncc.go.jp (K.U.); ttsuchid@ncc.go.jp (T.T.); 2Department of Respiratory Medicine, National Center for Global Health and Medicine, Tokyo 1628655, Japan; 3Department of Thoracic Oncology, National Cancer Center Hospital, Tokyo 1040045, Japan

**Keywords:** bronchoscopy, metastatic lung tumor, peripheral pulmonary lesion, radial endobronchial ultrasound

## Abstract

**Simple Summary:**

Metastatic lung tumors are relatively common, and their pathological diagnosis is crucial for determining the appropriate treatment strategy. Although bronchoscopy is one biopsy method, its role has not been established for peripheral metastatic lung tumors. The present study aimed to investigate the value of bronchoscopy, using radial endobronchial ultrasound for diagnosis. We analyzed 235 lesions consecutively, and the overall diagnostic yield was 76.6%, which was slightly higher than the reported cumulative sensitivity for general peripheral pulmonary lesions. There were no serious complications. The results demonstrated that bronchoscopy is a valuable technique for peripheral metastatic lung tumors that combines diagnostic accuracy and safety. Moreover, the higher diagnostic yield was associated with the large lesion size, inner location, and visibility on radiography. These findings should contribute to the selection of biopsy methods for metastatic lung tumors and improve the diagnostic yield.

**Abstract:**

Although lungs are one of the most frequent sites of metastasis for malignant tumors, little has been reported about the value of bronchoscopy for lung metastases presenting with peripheral pulmonary lesions (PPLs). This retrospective cohort study investigated the diagnostic value of bronchoscopy for peripheral metastatic lung tumors. Consecutive patients who underwent diagnostic bronchoscopy with radial endobronchial ultrasound for PPLs and were finally diagnosed with metastatic lung tumors from April 2012 to March 2019 were included. We analyzed 235 PPLs, with a median size of 18.8 mm. The overall diagnostic yield was 76.6%. In a multivariable analysis, large lesion size (>20.0 mm vs. <20.0 mm: 87.6% vs. 67.7%, *p* = 0.043, OR = 2.26), inner location (inner 2/3 vs. outer 1/3: 84.8% vs. 69.1%, *p* = 0.004, OR = 2.79), and visibility on radiography (visible vs. invisible: 83.2% vs. 56.1%, *p* = 0.015, OR = 3.29) significantly affected the diagnostic yield. Although a positive bronchus sign tended to have a higher yield, no significant difference was observed (81.8% vs. 70.6%, *p* = 0.063). Only one case of lung abscess was observed, with no serious complications. In conclusion, bronchoscopy is a valuable technique for peripheral metastatic lung tumors, with good diagnostic accuracy and safety.

## 1. Introduction

Metastasis is the leading cause of mortality in patients with cancer and is a topic of growing interest in clinical research. The lungs are one of the most frequent sites of metastasis for many malignant tumors, including the liver, brain, and bones, and a subset of patients lack any other evidence of disease [1]. In addition, metastatic lung tumors from a primary extrapulmonary malignancy are often an indicator of cancer progression [2].

To date, there has been a lack of evidence for defining appropriate diagnostic and therapeutic strategies for the development of new lung lesions in patients with known malignancies. For cases of isolated lung lesions, the pathological diagnosis of new lesions can help to distinguish between benign and malignant tumors, and primary lung cancer and metastatic lung tumors. Metastatic lung tumors are strongly suspected when multiple lung lesions are identified. Furthermore, pathological diagnosis is useful when searching for unknown primary tumors or for differentiating the second primary sites of multiple primary cancers [3,4]. Lung biopsy plays a key role in confirming staging and recurrence after surgery for lung metastasis of primary lung cancer.

The pathological diagnosis of lung lesions is crucial for determining the appropriate treatment strategy. For example, surgery is an option for a single metastatic lung tumor in colorectal cancer but can be controversial for multiple tumors [5]. Additionally, tissue samples are needed to predict the prognosis of patients with single or multiple tumors that cannot be operated on for various reasons and to determine the appropriate treatment strategy for chemotherapy. Recently, the detection of genetic mutations in malignant tumors (human epidermal growth factor receptor 2 in breast cancer [6] and v-Raf murine sarcoma viral oncogene homolog B in colorectal cancer [7]) and an immune analysis have become increasingly important, highlighting the utility of performing biopsies for new lung lesions whenever possible.

Biopsy techniques for lung lesions include surgical resection, transthoracic needle biopsy (TTNB), and bronchoscopy. Surgical resection is preferred when nonsurgical biopsies fail to diagnose lesions or highly suspected malignancies, based on radiological findings and patient characteristics. Nevertheless, surgery requires general anesthesia and is highly invasive. TTNB is less invasive than surgical resection, but its major limitation is the relatively high risk of complications, such as pneumothorax, air embolism, and dissemination of cancer cells into the pleural cavity [8,9].

Bronchoscopy is another biopsy technique that is well-established as a safe approach with a relatively low risk of complications. Lesions can be divided into endobronchial and peripheral pulmonary lesions (PPLs), which each have different diagnostic yields. The yield is higher for endobronchial lesions because they can be approached under direct observation via the scope; conversely, the yield is lower for PPLs for the opposite reason. Previously reported yields for metastatic lung tumors were relatively low, ranging from 50.0–72.6% [10,11,12]. However, these studies included a mix of both types of lesions; thus, the yield specific to PPLs remains unclear. Furthermore, the results were obtained before the widespread use of radial endobronchial ultrasound (R-EBUS).

The American College of Chest Physicians guidelines recommend the use of R-EBUS for diagnosing PPLs, since this approach improves the diagnostic yield [13]. Similar findings may be expected for peripheral metastatic lung tumors. A meta-analysis reported that the pooled diagnostic yield of bronchoscopy with R-EBUS for PPLs was 70.6% [14]. In particular, for cases of peripheral metastatic lung tumors, the yields were reported to be 66.7% (two out of three cases) [15] and 81.7% (26 out of 32 cases) [16]. Nevertheless, these reports were based on a small number of cases, and there is a paucity of research on the factors affecting the diagnostic results. Therefore, this study aimed to investigate the diagnostic value of bronchoscopy for peripheral metastatic lung tumors.

## 2. Materials and Methods

### 2.1. Patients

Consecutive patients who underwent diagnostic bronchoscopy with R-EBUS for PPLs at our institution between April 2012 and March 2019 were reviewed. Cases with a final diagnosis of metastatic lung tumors were included. If the cases were suspected of metastasis but could not be diagnosed using bronchoscopy, we referred to the results of other procedures (surgical biopsy, TTNB, and endobronchial ultrasound-guided transbronchial needle aspiration (EBUS-TBNA)). For cases that did not undergo these procedures, the final diagnosis was determined based on whether the lesion was treated as a metastatic lung tumor by referring to the treatment course and imaging follow-up. Patients with hematologic malignancies and rebiopsies of already diagnosed tumors for genetic profile analyses were excluded. The diagnostic yield was defined as the percentage of cases with a definitive diagnosis using bronchoscopy.

This study was approved by the National Cancer Center Institutional Review Board (No. 2018-090). The requirement for informed consent was waived due to the retrospective nature of the study.

### 2.2. Equipment and Procedures

Bronchoscopies were performed using a bronchoscope (BF-P260F, BF-P290, BF-1T260, BF-1TQ290, BF-Y0053, BF-Type260; Olympus, Tokyo, Japan) combined with an R-EBUS probe (UM-S20-17S or UM-S20-20S; Olympus, Tokyo, Japan). The equipment was selected by each operator according to the characteristics of the target lesions. All bronchoscopies were performed under local anesthesia with conscious sedation.

The R-EBUS probe, alone or covered with a guide sheath included in a guide sheath kit (K-201 or K-203; Olympus, Tokyo, Japan), was inserted through the working channel of the bronchoscope into the target bronchus, and the location was confirmed by X-ray fluoroscopy. The obtained R-EBUS findings were classified as “within”, “adjacent to”, or “invisible”, based on previous reports [17]. After scanning the target PPL, the probe was removed while maintaining its position of the bronchoscope, and the guide sheath if uesd, for subsequent sampling. The R-EBUS probe was reinserted as necessary in order to confirm its position. We defined the initial R-EBUS finding as the first detection before sampling and the best R-EBUS finding as the most delineated detection close to the lesion during the procedure.

Prior to bronchoscopy, CT scans (slice thickness of 0.5–5 mm) were performed at total lung capacity within 4 weeks of bronchoscopy in all patients. CT datasets for each patient were sent to a computer, and virtual bronchoscopic navigation (Ziostation2^®^, Ziosoft Ltd., Tokyo, Japan; or LungPoint^®^, Bronchus Ltd., Mountain View, CA, USA) was constructed prior to the procedures. Transbronchial needle aspiration (TBNA) was performed using an aspiration needle (NA-1C-1 or NA-2C-1; Olympus, Tokyo, Japan) to allow the needle to penetrate the bronchial wall towards the target for cases in which the initial R-EBUS finding was “adjacent to” or “invisible.”

### 2.3. Variables

Data on potential factors affecting a diagnostic yield were collected, including the age, sex, lesion size, lobe, location, bronchus sign, distance from the costal pleura, visibility on radiography, number of pulmonary lesions, and use of navigation, needle, and guide sheath.

Continuous values were binarized by referring to the medians. The size of each PPL was measured as the largest diameter on CT axial images, and the location was determined by dividing the distance from the hilum into three equal parts; the central two-thirds was defined as “inner”, and the rest was defined as “outer” [18]. The bronchus sign was defined as positive when a bronchus leading directly to the target lesion was observed on CT [19]. The distance from the costal pleura was measured as the shortest perpendicular length from the lateral border of the lesion. The number of pulmonary lesions was classified as either single or multiple at the time of bronchoscopy. All potential complications related to bronchoscopy were extracted.

### 2.4. Statistical Analyses

Descriptive statistics are presented as frequency percentages and numbers, and numeric data are presented as medians (ranges). Univariate analyses were performed using Fisher’s exact test or Pearson’s chi-square test, as appropriate. A multivariable analysis was performed using logistic regression to determine the factors associated with an increased diagnostic yield; variables with *p*-values < 0.20 in univariable analyses were selected and used. Two-tailed *p*-values < 0.05 were considered to indicate statistical significance. All statistical analyses were performed using JMP 14.2^®^ (SAS Institute Inc., Cary, NC, USA).

## 3. Results

In total, 235 PPLs in 234 patients were analyzed. Baseline characteristics of the lesions are presented in Table 1. The median size was 18.8 (range, 6.0–93.4) mm, and 130 lesions (55.3%) were < 20.0 mm. A total of 123 PPLs (52.3%) were located in the outer one-third, and most lesions were visible on radiography (178 PPLs, 75.7%). Moreover, the bronchus sign was positive in more than half of all cases (126 PPLs, 53.6%).

The overall diagnostic yield was 76.6% (Table 2). Of the remaining lesions, 38 were diagnosed by surgical biopsy, seven by TTNB, three by EBUS-TBNA, two by reattempting bronchoscopy, and one by biopsy of the axillary lymph node. Four patients without any pathological diagnoses were included because the clinical course was consistent with that of metastatic lung tumors. Various types of tumors were included, as summarized in Table 3. The most frequent tumor type was breast cancer, followed by colorectal and uterine cancers.

In the multivariable analysis, large lesion size (87.6% vs. 67.7%, *p* = 0.043), inner location (84.8% vs. 69.1%, *p* = 0.004), and visibility on radiography (83.2% vs. 56.1%, *p* = 0.015) significantly affected the diagnostic yield (Table 2). Although a positive bronchus sign tended to be associated with a higher yield, no significant difference was observed (81.8% vs. 70.6%, *p* = 0.063). The diagnostic yields for each of the best R-EBUS findings are listed in Table 4. The yields improved significantly as the R-EBUS moved closer to the lesion (within: 94.9%, adjacent to: 73.9%, and invisible: 13.0%; *p* < 0.001).

The relationship between the bronchus sign and the R-EBUS findings is described in Table 5. Even in lesions with positive bronchus signs, the initial R-EBUS finding was often “adjacent to” rather than “within” (46.8% and 41.3%, respectively). The position of the R-EBUS probe changed during the procedure and sometimes moved closer to the lesion than the initial R-EBUS findings before sampling. In particular, the R-EBUS findings were significantly improved when a needle was used, compared to when it was not used (29.0% vs. 12.5%, *p* = 0.002) (Table 6). Figure 1 shows a representative case in which the R-EBUS findings were improved after TBNA.

Of the 180 patients diagnosed by bronchoscopic biopsy, 95 were treated with chemotherapy, 6 with radiotherapy, 55 with surgery, and 24 with other methods (e.g., best supportive care, change of hospital). One patient presented with a lung abscess, but pneumothorax was not observed. In addition, only minor bleeding was observed in 16 patients, and no serious complications were identified.

## 4. Discussion

In the present study, we evaluated the diagnostic value of bronchoscopy using R-EBUS for peripheral metastatic lung tumors. To the best of our knowledge, this is the first study to systematically analyze the diagnostic yield in this field. The overall diagnostic yield was 76.6%, which was slightly higher than the reported cumulative sensitivity of bronchoscopy for PPLs using R-EBUS [14]. The diagnostic yield for metastatic lung tumors varies by cancer type, with breast, colorectal, and renal cancers were reported to have a higher diagnostic yield due to their higher affinity to the bronchus [20,21]. In this regard, the results of this study exhibited a similar trend (Table 3).

We identified the larger lesion size, inner location, and radiography visibility as factors that increased the diagnostic yield. Previous reports have indicated that the yields in these groups on bronchoscopy were high regardless of whether R-EBUS was employed [13,18]. A meta-analysis of bronchoscopy with R-EBUS reported that the lesion size (> 2 cm), malignant nature, and a positive bronchus sign on CT were associated with a higher diagnostic yield [14]. The current findings suggest that the factors influencing the diagnostic yield for PPLs are similar for both primary and metastatic lung tumors, with the exception of the bronchus sign.

Although the bronchus sign has been reported as a powerful predictive factor for successful transbronchial biopsy for PPLs in numerous studies [14,17,22,23], we did not observe a significant difference in this study (positive vs. negative: 81.8% vs. 70.6%, *p* = 0.063). Typically, a positive bronchus sign is strongly correlated with diagnostic yield via an increased detection of the best R-EBUS findings [24]. In our study, 54% of lesions with a positive bronchus sign were classified as “within” in the best R-EBUS findings (Table 5), which contrasts the previously reported percentage of 70% [25,26], indicating a poor correlation between the bronchus sign and R-EBUS findings in metastatic lung tumors. We hypothesize that the cause of this discrepancy may be related to the pathogenesis of metastatic lung tumors. Metastatic lesions often do not reach the bronchi because cancer cells generally break away from the primary lesion via the blood or lymph flow and form metastatic tumors [27]. Thus, there may be cases with a pseudo-positive bronchus sign in which metastatic lesions did not involve the bronchi but only compressed them, despite the appearance of a positive bronchus sign on CT.

Another potential factor contributing to this discrepancy is the influence of changes in the R-EBUS findings during the procedure (Table 5). The bronchus sign and diagnostic yield were more strongly associated with the initial R-EBUS findings and best R-EBUS findings, respectively. For PPLs lacking the bronchus sign, TBNA has been reported to be more effective than forceps biopsy [28,29]. Although the indication for TBNA should ultimately be determined with reference to the R-EBUS findings [30], discrepancies between the bronchus sign and the R-EBUS findings in metastatic lung tumors may exist, as mentioned above. In this regard, for cases in which the R-EBUS did not reach the inside of the lesion, we performed TBNA as much as possible, which significantly improved the R-EBUS findings (Table 6). In addition, cryobiopsy has been reported to be effective for PPLs for which the R-EBUS is indicated adjacent to the lesion [31]. Collectively, these findings highlight the potential utility of cryobiopsy and TBNA for diagnosing metastatic lung tumors.

The diagnostic yield and its influencing factors for metastatic lung tumors were similar to those of PPLs, indicating that bronchoscopy should be considered an appropriate biopsy technique for metastatic lung tumors. In particular, although TTNB and surgical biopsy are more invasive for centrally located lung lesions, the diagnostic yield of bronchoscopy was high, suggesting that bronchoscopy is a superior diagnostic technique for metastatic lung tumors located in the inner two-thirds. TTNB has a diagnostic accuracy of over 90% and is often more effective than bronchoscopy for diagnosing PPLs [32] but requires penetration of the pleura and carries a high risk of pneumothorax. In this regard, the risk of pneumothorax increases with the depth of the distance from the pleura [33]. No cases of pneumothorax were observed in our study, highlighting the low risk of complications from bronchoscopy and the diagnostic utility of this approach.

Surgical biopsy is the most reliable diagnostic technique, with almost 100% sensitivity and specificity, and it can be curative in a subset of malignancies. The distinct advantage of surgical biopsy is that diagnosis, staging, and therapy are performed in a single operative procedure. However, the surgical method differs depending on whether it is primary lung cancer or isolated lung metastasis: the former requires a lobectomy with lymph node dissection, whereas the latter requires partial resection. When the intraoperative diagnosis is initially non-definitive and the histopathology is definitive for non-small cell lung cancer, a second surgery may be required for treatment. In addition, several factors have been reported to play key roles in the outcomes of lung metastasectomy, including tumor type and histology, number and size of lung metastases, location, disease-free interval, completeness of resection, and surgical approach [34,35,36,37]. Therefore, preoperative diagnosis is important in the selection of surgical candidates and appropriate approaches. In this regard, bronchoscopy may play a major role due to its combination of diagnostic accuracy and safety. In fact, of the 93 patients who underwent surgery in this study, 55 had preoperative diagnoses of metastatic lung tumors by bronchoscopy, and the appropriate surgical method was selected, while some patients who were treated with other therapies were able to avoid surgery based on the bronchoscopic diagnosis.

This study has several limitations. First, it was a retrospective, non-randomized study in a single institution. As such, there may have been selection bias, whereby patients with a diagnosis of metastatic lung tumors were enrolled. We believe that there is little significant bias in the selection for bronchoscopic biopsy in terms of diagnostic difficulty, because bronchoscopic biopsy is usually the first approach for new lung lesions at our institution. Second, the procedures were not performed by the same bronchoscopist, and the type of bronchoscope and sampling device (forceps, brush, and/or needle) were selected independently by each bronchoscopist. Third, not all patients were diagnosed with pathology. The differences would likely have been minimal if patients without a final pathological diagnosis were excluded, given that only four such cases were identified in this study. Nevertheless, further multicenter prospective cohort studies are warranted to verify our findings.

## 5. Conclusions

Bronchoscopy is a valuable diagnostic technique for peripheral metastatic lung tumors, which combines diagnostic accuracy and safety. The lesion size, location, and radiography visibility of the lesion are important in selecting its indication for bronchoscopy, because they affect its diagnostic yield. In addition, it should be noted that the bronchus may not reach the tumor, despite a positive bronchus sign in some cases of metastatic lung tumors.

## Figures and Tables

**Figure 1 cancers-14-00375-f001:**
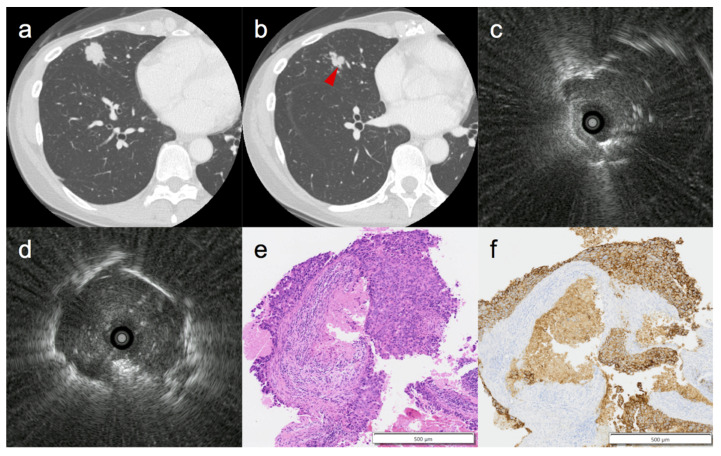
Representative case of a 48-year-old female for whom needle aspiration was effective. (**a**) High-resolution computed tomography showing a solid nodule 19.8 mm in size in the middle lobe. (**b**) Proximal side of the lesion indicating a positive bronchus sign (arrowhead). (**c**) Initial radial endobronchial ultrasound (R-EBUS) findings showing “adjacent to” the lesion. (**d**) After needle aspiration, the R-EBUS finding changed to “within” the lesion. (**e**) Hematoxylin and eosin staining (×40) depicting adenocarcinoma. (**f**) Positive immunohistochemical staining for ERBB2 (×40) similar to the primary lesion, with a diagnosis of metastasis of breast cancer.

**Table 1 cancers-14-00375-t001:** Baseline characteristics of lesions.

Variable	N (%) or Median (Range)
Age (years)	66 (27–86)
≤65	109 (46.4)
>65	126 (53.6)
Sex	
Male	112 (47.7)
Female	123 (52.3)
Lesion size (mm)	18.8 (6.0–93.4)
Small (≤20)	130 (55.3)
Large (>20)	105 (44.7)
Lobe	
Upper	100 (42.5)
Middle or lingular	42 (17.9)
Lower	93 (39.6)
Distance from the costal pleura (mm)	14.2 (0–65.0)
Near (≤10)	99 (42.1)
Far (>10)	136 (57.9)
Location	
Outer 1/3	123 (52.3)
Inner 2/3	112 (47.7)
Bronchus sign	
Positive	126 (53.6)
Negative	105 (44.7)
Visibility on radiography	
Visible	178 (75.7)
Invisible	57 (24.3)
Number of pulmonary lesions	
Single	116 (49.4)
Multiple	119 (50.6)
Navigation	
Used	204 (86.8)
Not used	31 (13.2)
Needle	
Used	107 (45.5)
Not used	128 (54.5)
Guide sheath	
Used	146 (62.1)
Not used	89 (37.9)

**Table 2 cancers-14-00375-t002:** Univariable and multivariable analyses of factors associated with diagnostic yield.

Variable	Diagnostic Cases, N (%)	Nondiagnostic Cases, N (%)	Univariable	Multivariable	
			*p*-Value	*p*-Value	Adjusted OR (95% CI)
Overall	180 (76.6)	55 (23.4)	-	-	-
Age (years)			0.122	0.228	1.54 (0.76–3.14)
≤65	78 (71.6)	31 (28.4)			
>65	102 (81.0)	24 (19.1)			
Sex			1.000	0.771	1.11 (0.54–2.27)
Male	86 (76.8)	26 (23.2)			
Female	94 (76.4)	29 (23.6)			
Lesion size (mm)			< 0.001	0.043	2.26 (1.01–5.04)
Small (≤20)	88 (67.7)	42 (32.3)			
Large (>20)	92 (87.6)	13 (12.4)			
Lobe			0.398	NA	NA
Upper	79 (79.0)	21 (21.0)			
Middle or lingular	34 (81.0)	8 (19.0)			
Lower	67 (72.0)	26 (28.0)			
Distance from the costal pleura (mm)			0.086	NA	NA
Near (≤10)	70 (70.7)	29 (29.3)			
Far (>10)	110 (80.9)	26 (19.1)			
Location			0.005	0.004	2.79 (1.36–5.70)
Outer 1/3	85 (69.1)	38 (30.9)			
Inner 2/3	95 (84.8)	17 (15.2)			
Bronchus sign			0.063	0.207	1.58 (0.77–3.22)
Positive	103 (81.8)	23 (18.3)			
Negative	77 (70.6)	32 (29.4)			
Visibility on radiography			< 0.001	0.015	3.29 (1.57–6.91)
Visible	148 (83.2)	30 (16.9)			
Invisible	32 (56.1)	25 (43.9)			
Number of pulmonary lesions			0.090	0.148	1.64 (0.84–3.22)
Single	83 (71.6)	33 (28.5)			
Multiple	97 (81.5)	22 (18.5)			
Navigation			0.254	NA	NA
Used	159 (77.9)	45 (22.1)			
Not used	21 (67.7)	10 (32.3)			
Needle			0.359	NA	NA
Used	85 (79.4)	22 (20.6)			
Not used	95 (74.2)	33 (25.8)			
Guide sheath			0.017	NA	NA
Used	104 (71.2)	42 (28.8)			
Not used	76 (85.4)	13 (14.6)			

OR, odds ratio; CI, confidence interval; NA, not applicable.

**Table 3 cancers-14-00375-t003:** Tumor types and associated diagnostic yield.

Tumor Type	Diagnostic Cases, N (%)
Breast cancer	46/52 (88.5)
Colorectal cancer	33/38 (86.8)
Uterine cancer	15/23 (65.2)
Pancreatic cancer	8/13 (61.5)
Lung cancer	7/13 (53.8)
Oral cancer	7/10 (70.0)
Gastric cancer	7/8 (87.5)
Esophageal cancer	7/8 (87.5)
Laryngeal cancer	6/8 (75.0)
Urothelial cancer	6/7 (85.7)
Biliary tract cancer	6/7 (85.7)
Renal cancer	5/7 (71.4)
Prostate cancer	5/6 (75.0)
Melanoma	4/4 (100.0)
Liposarcoma	1/4 (25.0)
Uterine leiomyosarcoma	2/3 (66.7)
Thymic cancer	2/3 (66.7)
Sarcoma, NOS	0/3 (0.0)
Solitary fibrous tumor	2/2 (100.0)
Uterine carcinosarcoma	2/2 (100.0)
Epithelioid sarcoma	2/2 (100.0)
Chondrosarcoma	2/2 (100.0)
Hepatocellular cancer	2/2 (100.0)
Ovarian cancer	1/2 (50.0)
Skin cancer	1/1 (100.0)
Angiosarcoma	1/1 (100.0)
Osteosarcoma	0/1 (0.0)
Testicular sarcoma	0/1 (0.0)
Follicular dendritic cell sarcoma	0/1 (0.0)
Germ cell tumor	0/1 (0.0)

NOS, not otherwise specified.

**Table 4 cancers-14-00375-t004:** Diagnostic yield according to R-EBUS findings.

R-EBUS Finding	Diagnostic Cases, N (%)	*p*-Value
Within the lesion	92/97 (94.9)	<0.001
Adjacent to the lesion	85/115 (73.9)	
Invisible lesion	3/23 (13.0)	

R-EBUS, radial endobronchial ultrasound.

**Table 5 cancers-14-00375-t005:** Relationship between bronchus sign and R-EBUS findings.

Bronchus Sign	Initial R-EBUS Finding			Best R-EBUS Finding		
	Within the Lesion, N (%)	Adjacent to the Lesion, N (%)	Invisible Lesion, N (%)	Within the Lesion, N (%)	Adjacent to the Lesion, N (%)	Invisible Lesion, N (%)
Positive	52 (41.3)	59 (46.8)	15 (11.9)	68 (54.0)	47 (37.3)	11 (8.7)
Negative	10 (9.2)	75 (68.8)	24 (22.0)	29 (26.6)	68 (62.4)	12 (11.0)

R-EBUS, radial endobronchial ultrasound.

**Table 6 cancers-14-00375-t006:** Additional univariable analysis of changes in R-EBUS findings using TBNA.

	Improved *, N (%)	Not Improved, N (%)	*p*-Value
Needle			0.002
Used	31 (29.0)	76 (71.0)	
Not used	16 (12.5)	112 (87.5)	

* The breakdown of cases with improved R-EBUS findings is as follows: from “adjacent to” to “within”, 31 cases; from “invisible” to “within”, 4 cases; and from “invisible” to “adjacent to”, 12 cases.

## Data Availability

The data published in this study are available upon reasonable request from the corresponding author. To obtain date, please send us an e-mail.

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
