# Peer review of "Diagnostic Value of Bronchoscopy for Peripheral Metastatic Lung Tumors"

_cancers, 2022, doi:10.3390/cancers14020375_

Round 1
Reviewer 1 Report
It is very nice study. I have a few questions. First, what patients did you do this procedure? I mean, is every patient with metastatic lung lesion or every patient with lung lesion? In regular setting, bronchoscopy biopsy is applied for those has chance to get proof. The yield rate is about 76%, which is very high but it’s related with how you select the subjects. Second, you mention the importance of this procedure is mainly on the difference of the following treating methods. (Primary vs second lung cancer) I am curious about your overall experience. How much did you change the following treating strategy? What is the benefits of this additional procedure compared with upfront resection?
Author Response
Thank you for providing the valuable questions.
First answer.
Our institution specializes in oncology, and our policy is to make a pathological diagnosis whenever possible and provide appropriate treatment when new lesions appear in patients. The lungs are one of the most frequent sites of metastasis for many malignant tumors, and biopsy is often considered in the following cases as summarized in the Introduction: (1) new lung lesions in patients with known malignancies; (2) when searching for unknown primary tumors or for differentiating the second primary sites of multiple primary cancers; and (3) confirming staging and recurrence after surgery for lung metastasis of primary lung cancer. Patients with new lung lesions are often referred for bronchoscopic biopsy as transthoracic biopsy is not actively performed due to concerns about serious complications.Even in patients who are considered for surgery, bronchoscopy is often requested in advance to determine the indications for curative surgery. The choice of these biopsy methods is primarily made by the attending physicians, and we have not usually intervened. Therefore, we believe that there is little significant bias in the selection for bronchoscopic biopsy in terms of diagnostic difficulty.
To clarify this view, we have added the following text to the limitation.
In lines 317–320 on page 12: “We believe that there is little significant bias in the selection for bronchoscopic biopsy in terms of diagnostic difficulty because bronchoscopic biopsy is usually the first approach for new lung lesions at our institution.”
Second answer.
Among the 93 patients who underwent surgery in this study, 55 were diagnosed with metastatic lung tumors by bronchoscopic biopsy in advance. As described in lines 297–309 on page 11, the surgical method differs depending on whether it is primary or metastatic. The former requires lobectomy with lymph node dissection, and the latter requires partial resection. Although upfront resection depends on the intraoperative pathological diagnosis to modify the surgical procedure, rapid pathological diagnosis has the problem of decreasing reliability; thus, preoperative diagnosis is more beneficial. Actually, the 55 patients underwent appropriate surgical procedures based on their preoperative diagnosis. Furthermore, the treatment plan included 95 chemotherapy, 6 radiotherapy, 55 surgery, and 24 others (e.g., BSC, change of hospital), and surgery itself could be avoided in part of them.
We have added the sentences below to reflect the contents in lines 237–239 on page 10 and lines 310–314 on page 11, respectively.
“Of the 180 patients diagnosed by bronchoscopic biopsy, 95 were treated with chemotherapy, 6 with radiotherapy, 55 with surgery, and 24 with others (e.g., best supportive care, change of hospital).”
“In fact, of the 93 patients who underwent surgery in this study, 55 had a preoperative diagnosis of metastatic lung tumor by bronchoscopy and the appropriate surgical method was selected, and part of patients who were treated with other therapies were able to avoid surgery based on the bronchoscopic diagnosis.”

Reviewer 2 Report
Review comments
Diagnostic value of bronchoscopy for peripheral metastatic lung tumors
There has been increasing need for pathological confirmation of metastatic lung tumor in daily clinics. This report provides many useful information not only for bronchoscopists but also for pulmonologists or lung cancer specialists.
I have only some minor comments.
- CT bronchus sign is sometimes difficult to determine positive or negative. As authors mentioned, bronchus sign might be false positive in many metastatic lesions. How did authors decide CT bronchus sign positivity? If bronchus was detected just adjacent to the lesions, the bronchus sign is positive or negative?
- Data in table 4-6 and figure 1 were interesting but should be mentioned in Result section. Only interpretation or discussion of these data should be in the Discussion.
- In table 6, even in patients who did not receive additional TBNA, EBUS findings were improved in 12.5%. How did authors improve EBUS findings?
Author Response
- CT bronchus sign is sometimes difficult to determine positive or negative. As authors mentioned, bronchus sign might be false positive in many metastatic lesions. How did authors decide CT bronchus sign positivity? If bronchus was detected just adjacent to the lesions, the bronchus sign is positive or negative?
Thank you for your important comment. To the best of our knowledge, there is no standardized criteria for positive or negative on the bronchus sign. At our institution, we determine the bronchus sign based on the Tsuboi classification, as reported by Gaeta et al., and we followed it in this study. Specifically, it was defined as positive when the bronchial lumen was obstructed by the tumor (Tsuboi type 1) or contained by the tumor (Tsuboi type 2). Negative bronchus sign was defined as the bronchial lumen being compressed by the tumor (Tsuboi type 3) or becoming invisible before reaching the tumor by peribronchial or submucosal spread of the tumor or by the enlarged nodes (Tsuboi type 4). Therefore, if bronchus was detected just adjacent to the lesions, the bronchus sign was determined to be negative.
To clarify the source of the bronchus sign, the following reference has been added. The citation numbers have been changed throughout to reflect this.
As number 19 in the References: “Gaeta, M.; Pandolfo, I.; Volta, S.; Russi, E. G.; Bartiromo, G.; Girone, G.; La Spada, F.; Barone, M.; Casablanca, G.; Minutoli, A. Bronchus sign on CT in peripheral carcinoma of the lung: value in predicting results of transbronchial biopsy. A.J.R. Am. J. Roentgenol. 1991;157(6):1181–1185.”
- Data in table 4–6 and figure 1 were interesting but should be mentioned in Result section. Only interpretation or discussion of these data should be in the Discussion.
Thank you for your suggestion. We have added the sentences below to reflect the contents of tables 4–6 and figure 1.
In lines 209–217 on page 8: “The relationship between the bronchus sign and R-EBUS findings is described in Table 5. Even in lesions with positive bronchus signs, the initial R-EBUS finding was often “adjacent to” rather than “within” (46.8% and 41.3%, respectively). The position of the R-EBUS probe changed during the procedure and got sometimes closer to the lesion than the initial R-EBUS findings before sampling. In particular, the R-EBUS findings were significantly improved when a needle was used compared to when it was not used (29.0% vs. 12.5%, P = 0.002) (Table 6). Figure 1 shows a representative case in which the R-EBUS findings were improved after TBNA.”
- In table 6, even in patients who did not receive additional TBNA, EBUS findings were improved in 12.5%. How did authors improve EBUS findings?
Thank you for your meaningful question. Of the 16 patients with improved R-EBUS findings without additional TBNA, shown in table 6, 13 patients had been performed brushing, and all had been performed forceps biopsy. It is sometimes experienced that brushing or forceps biopsy improves R-EBUS findings even without TBNA. In such cases, we assumed that the bronchial lumen may have been narrowed in front of the tumor, as in Tsuboi type 4, and that sampling may have opened it, resulting in improved R-EBUS findings. In any case, it is clear that R-EBUS findings were more frequently improved with TBNA.
